# The association between the rise of gun violence in popular US primetime television dramas and homicides attributable to firearms, 2000–2018

**Patrick E. Jamieson, Daniel Romer** *

Annenberg Public Policy Center, University of Pennsylvania, Philadelphia, PA, United States of America

* dan.romer@appc.upenn.edu

## Abstract

Injuries and fatalities due to firearms are a major burden on public health in the US. The rise in gun violence in popular movies has been suggested as a potential cultural influence on this behavior. Nevertheless, homicide rates have not increased over recent decades in the US, suggesting that media portrayals have had little influence on gun violence. Here we challenge this interpretation by examining trends in the proportion of violence that are attributable to firearms, a measure that should be more sensitive to media violence. In addition, we examine trends in the portrayal of guns in popular television (TV) dramas, which are viewed more frequently than movies. We ask (a) whether gun violence has increased in these TV shows not only on an absolute basis but also as a proportion of violent scenes and (b) whether trends in gun portrayal on these shows are associated with corresponding trends in the proportion of real-world violence attributable to firearms in the US from 2000 to 2018. To answer these questions, we coded annual instances of violence, gun violence, and proportion of violence involving guns for each 5-minute segment of 33 popular TV dramas in the police, medical, and legal genres from 2000 to 2018. Trends in annual rates of violence, gun violence and proportion of violence involving guns were determined over the study period and were compared to annual rates of homicide attributable to firearms in three age groups: 15–24, 25–34 and 35 and older. Although violence on TV dramas peaked in 2011, gun use steadily increased over the study period both in absolute terms and in relation to other violent methods. The latter metric paralleled trends in homicides attributable to firearms for all three age groups, with the strongest relationship for youth ages 15–24 ($R^2 = .40$, $P = .003$). The positive relation between relative amount of TV violence involving guns and actual homicides due to firearms, especially among youth, is consistent with the hypothesis that entertainment media are contributing to the normative acceptance of guns for violent purposes. Future research is needed to study the influence of media violence on gun acquisition at the individual level.

**Data Availability Statement:** All relevant data are within the paper and its Supporting Information file.

**Funding:** The Robert Wood Johnson Foundation supported this research with a grant to the Annenberg Public Policy Center, Patrick Jamieson, PI. The funders had no role in the study design, data collection and analysis, decision to publish, or preparation of the manuscript.

**Competing interests:** The authors have declared no competing interests exist.

## Introduction

Injuries due to gun violence are increasingly seen as a public health crisis in the US [1]. In 2018, over 39,000 Americans died from gun-related deaths [2], along with over 70,000 non-fatal injuries attributable to firearms [3]. Young persons ages 15–24 have the highest rates of firearm homicides [2] and are especially sensitive to media influences that place them at risk of adverse health consequences, such as violence [4, 5]. This longstanding finding has drawn attention to the rise in lethal violence [6] and the use of guns for violent purposes in top grossing movies rated appropriate for adolescents [7, 8], raising the hypothesis that such portrayals normalize the use of guns, especially among youth vulnerable to violence [6, 7]. Indeed, even parents see movie depictions of gun use as acceptable for viewing by adolescents over the age of 14 if the violence is viewed as justified for the defense of self or others and does not display the upsetting effects of the violence [9]. Such depictions are not only attention-getting but are also seen as acceptable by adolescents [10], and even short-term laboratory exposure to guns in movies enhances interest in their use by pre-adolescents [11]. Thus, there is evidence that the use of guns for self-defense and other ethically acceptable forms of violence in entertainment could be a source of imitation, especially in youth vulnerable to such influence.

The hypothesis that the increasing portrayal of gun use in entertainment media has influenced young people has been questioned because homicides among youth in the US have actually declined over the past several decades [12, 13]. While this may partly be the result of better trauma care in saving the lives of some gunshot victims [14], *rates of homicide are not the critical outcome to assess the media influence hypothesis.* If media depictions make the use of guns more salient and acceptable, then one would expect their use as weapons to increase relative to other means of violence. Thus, even if homicide rates have declined in recent decades, the use of guns as a proportion of all homicides may have increased. Because this outcome is more appropriate to test hypotheses about media influence, we focus on it in the present study.

We also focus on a form of media entertainment that is even more heavily viewed by adolescents than movies [15, 16], namely popular television (TV) dramas. The same form of bloodless violence that is common in popular movies is also common on TV dramas [17]. However, less is known about recent trends in gun violence on TV. Violence in popular TV dramas declined from the 1970's to the 1990's, but this trend began to reverse in 2000 [18]. Here we present findings regarding the presence of violence and the use of guns for violent purposes in popular TV dramas from 2000 to 2018. We anticipated that just as gun use has increased in popular movies, the same was true for entertainment on TV. We also hypothesized that if TV gun use serves to normalize the use of guns for violent purposes, this influence should be most apparent in the use of guns versus other violent methods on both TV and in the real world. Thus, we compared trends in the share of violent TV scenes in which guns were used with trends in the share of homicides attributable to firearms. The latter measure of firearm homicides has received less research attention but is the index that should be most sensitive to any normative influence of TV portrayal of gun use.

## Methods

Prime-time network dramas from among the top-30 of each year as ranked by Nielsen were sampled from 2000 to 2018. We selected 33 shows that were highly ranked over multiple seasons, with 60% of the shows in police, 33% in medical, and 21% in legal genres (see S1 Appendix). Shows are defined by season which often span adjacent years. We assigned shows to the year of their airing rather than season. The shows were coded on an average of 6.3 years over the course of the study period resulting in a dataset with 211 cases. All of the shows were rated as TV-14, which is defined as containing material that might be "unsuitable for children under

14-years-of age" [19]. We selected every other episode for coding, representing approximately 1,476 commercial-free-hours over the study period.

## Media coding

Trained research assistants coded every 5-minute segment of each show's episodes for the presence of violence and the use of firearms in those segments [20]. To begin coding, research assistants had to reach an inter-coder reliability $\geq$ .80 using Krippendorf's Alpha (K$\alpha$) [21]. The presence of *any violence* (0 = no; 1 = yes) was defined as "physical acts where the aggressor makes or attempts to make some physical contact with the intention of causing injury or death" (K$\alpha \geq$ .84) [22]. Each segment with *any violence* was further coded for *gun violence*, with gun defined as a "weapon that fires a bullet or energy beam with the intention of coercing or harming others." *Gun violence* was defined as "a gun was fired and also an animate being was hit"[5] (0 = no, 1 = yes) (K$\alpha \geq$ .90).

## Trend analysis

As a preliminary analysis, we conducted a multilevel mixed-effects regression to verify that the sum of the segments for each of the three codes for each of the 33 shows per year was related to trends over time controlling for the number of segments that were coded and for random intercepts of shows. This analysis indicated that both violence and gun violence contained reliable linear (*P*'s = .003, .006) and quadratic components (*P*'s = .019, .019). The occurrence of gun violence controlling for the number of violent segments only contained a linear component (*P* = .085). These analyses verified that time trends of the three codes were robust to controls for differences by show.

To determine annual rates, we calculated the percentage of segments for each show for each of the three metrics (violence, gun use, and proportion of violence using guns). To reduce the influence of outliers, we averaged the log transforms of the percentages greater than zero and converted those means back to the percentage scale. Linear regression was then used to identify annual trends over the study period. All analyses were conducted using Stata, version 15.

## Homicides in the US

We calculated the proportion of homicides attributable to firearms for three age groups (15–24, 25–34, and 35+) using national rates for each year in the study period [2]. Although the rates were adjusted to year 2000 population, the proportions we calculated should minimize differences in population size over time. To reduce ceiling effects, we transformed the proportions to the log of the corresponding odds ratio.

## Results

Annual percentages of TV violence increased from 22.4 in 2000 to 41.4 in 2011 before declining to 27.1 in 2018 (Fig 1A). There was both a linear (B = .66, 95% CI: .39, .93; *P* < .001) and quadratic trend (B = -.12, 95% CI: -.18, -.07; *P* < .001) over the study period.

Gun violence rose from 4.5% in 2000 to 9.0% in 2018 (Fig 1B), with both a linear (B = .44, 95% CI: .33, .55; *P* < .001) and quadratic trend (B = -.03, 95% CI: -.05, -.008; *P* = .012).

The percentage of violent segments that contained gun violence rose linearly from 21.1 in 2000 to 33.3 in 2018 (Fig 1C), B = .47, 95% CI: .20, .74; *P* = .002. Although the quadratic term did not reach significance, it was notably in the opposite direction from the quadratic trends in the other TV metrics, B = .05, 95% CI: -.005, .106; *P* = .072.

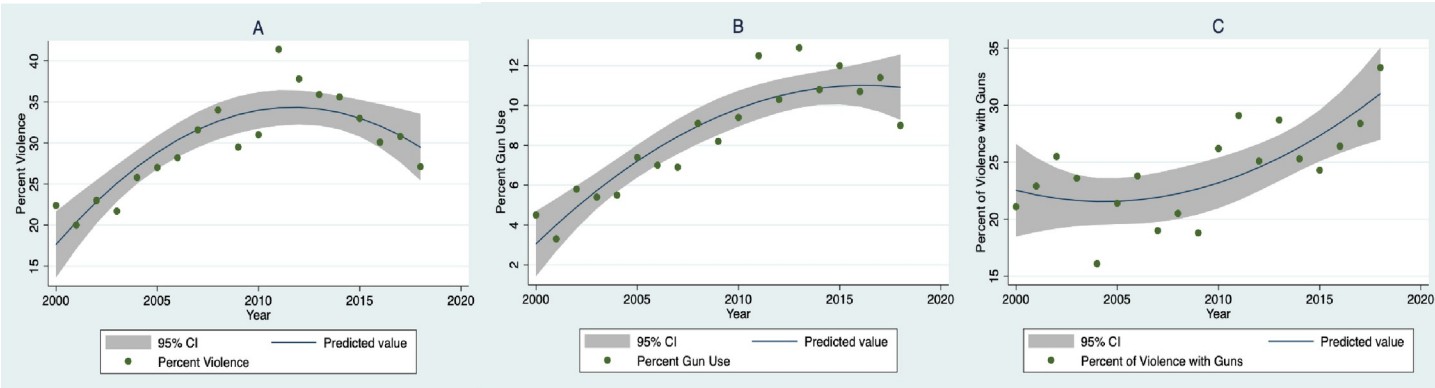

**Fig 1.** Plots and best fitting trends between study year and percent of TV segments with (A) violence, (B) gun use, and (C) gun use as percent of violence for years 2000 to 2018.

Fig 2 shows best fitting curves for the relation between year and rates of gun homicides for three age groups. The relation for youth ages 15 to 24 (Fig 2A) was primarily negative over the time period, B = -.077, 95% CI: -.147, -.008; *P* = .032, indicating that despite the rise in gun violence in TV dramas, the rate of gun homicides in this young age group was opposite to the TV trend.

The relation for persons in the 25 to 34 age range (Fig 2B) was mainly flat, with no linear or quadratic trends, *P*'s = .40, .36 respectively. For those ages 35 and older, there was both a linear and quadratic trend, indicating that gun homicides increased over the period (B = .017, 95% CI: .004, .030; *P* = .014) and also declined before rising (B = .005, 95% CI: .003, .008; *P* = .001). In sum, the time trends for firearm homicides ranged from negative to positive depending on the age range of the population.

Fig 3 has the log odds of firearm fatalities in the US over the study period for three age groups. As is evident, the trend was upward for all three groups and decidedly higher for the youngest. There was also a large dip in 2001 especially for persons older than 24, which was likely due to the large number of deaths resulting from the terrorist attack on September 11 and which mainly affected older persons. Because of this departure from trend, we first removed the variation due to this outlier with a dummy variable for that year and then added the relevant TV metric to determine its unique contribution to the equation.

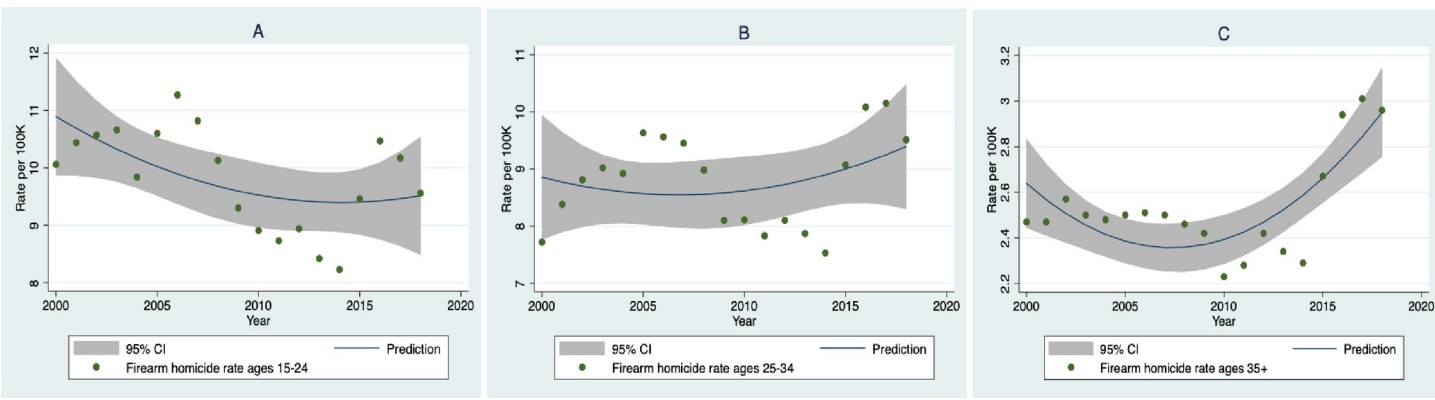

**Fig 2. Plots of best fitting trends between study year and firearm homicides for three age groups.**

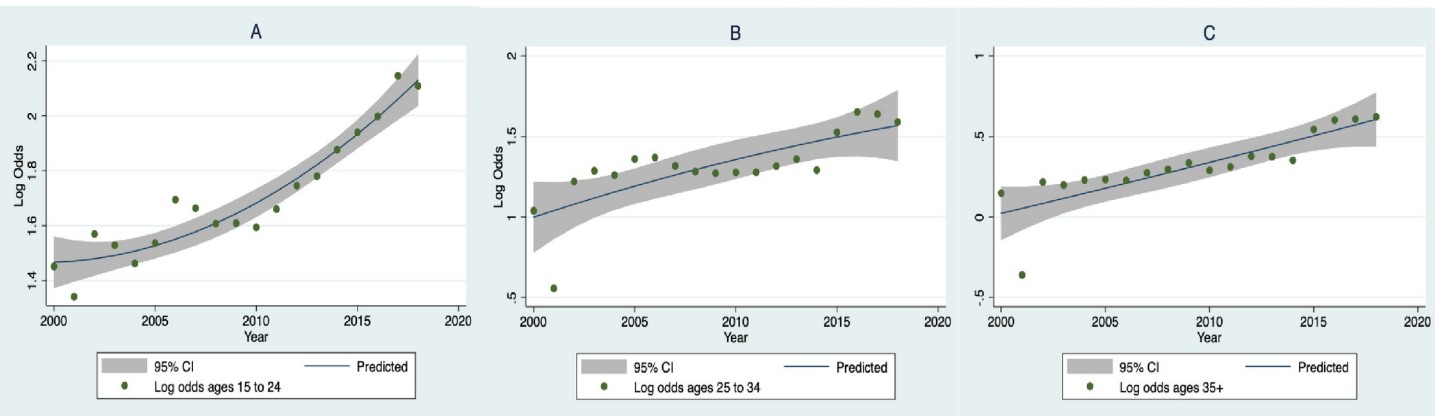

**Fig 3.** Plots and best fitting trends between study year and the log odds of homicides attributable to firearms for (A) ages 15 to 24, (B) ages 25 to 34, and (C) ages 35+.

Fig 4 displays the log odds of homicides attributable to firearms for three age groups plotted against the annual percentage of gun use within violent segments. The log odds of firearm homicides increased in parallel with TV gun use as a percentage of violent segments, especially in the 15–24 age group, which had the largest linear relation, B = .034, 95% CI: .014, .053; *P* = .002, and accounted for the largest share of variation across the age groups after removing the predictor for the abnormal year 2001: 40% vs. 11% for ages 25–34 and 16% for ages 35+. The linear relation between firearm homicides and TV gun use for the 25–34 age group was about 56% of the size of the 15 to 24 age group (B = .019, 95% CI: .002, .036; *P* = .028) and only slightly more for the 35+ age group, B = .021, 95% CI: .006, .036; *P* = .009.

As a robustness check, we added the average number of segments coded per year as a covariate. However, this potential confound did not add any prediction to the models, all *P*'s >.400.

The absolute measure of TV gun violence was also related to the three homicide outcomes, accounting for 37% of the variation in ages 15–24, 12% in ages 25–34, and 20% in ages 35+. The measure of overall violence, which peaked rather than steadily increasing across the time period, was weakly related to the homicide outcomes accounting for less than 10% of the variance in any age group.

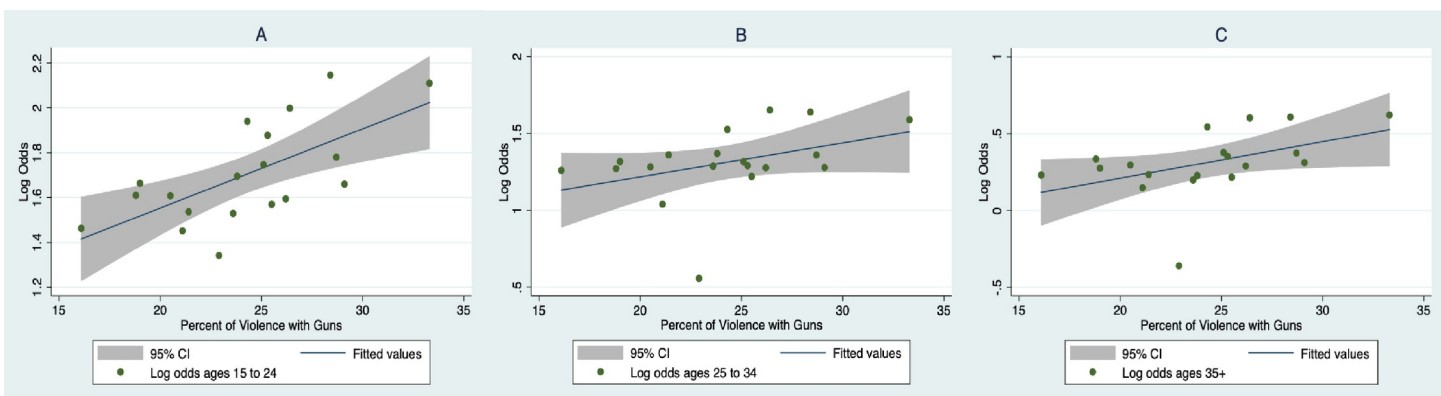

**Fig 4.** Plots and best fitting linear relations between percentages of gun use in violent TV segments and log odds of homicides attributable to firearms for persons (A) ages 15–2, (B) ages 25–34, and (C) ages 35+ from 2000 to 2018.

## Discussion

As has been found for popular movies [7, 8], the rate of gun violence has increased in popular TV dramas since 2000, not only in absolute terms, but importantly as a proportion of the violence on these programs. The increase in TV gun violence was predictive of firearm violence for all age groups, not only as measured by its absolute prevalence but also by its proportion of violent scenes, which is a theoretically more relevant measure since gun violence could increase without it being more prevalent than other methods. Furthermore, TV violence alone is an unlikely source of the association since it did not continue unabated over the study period and was less related to trends in firearm homicides. Thus, our measure of the relative use of guns is the most theoretically stringent measure of the cultural influence of entertainment media on this behavior.

The findings support the hypothesis that the prominent portrayal of guns in popular entertainment increases their adoption for violent purposes, a relation especially apparent in youth. The use of guns in popular movies is seen as justified by parents when it is used in defense of self or others [23], and similar portrayals are likely in TV dramas where police are heavily featured [17]. Youth are also accepting of such portrayals [10]. The trend in TV gun violence most closely paralleled the log odds of homicides attributable to firearms in young people ages 15 to 24 who are also most likely to be victims of firearm homicide.

We believe we are the first to study trends in the proportion of homicides attributable to firearms, especially for different age groups. Our results show that the use of firearms as a method of lethal violence in comparison to other methods has increased for all age groups in the US, indicative of a population-wide increase in this behavior. It is noteworthy that the increase in the use of firearms in homicides is not the result of greater violence over the study period. Homicides have declined for youth ages 15–24 and have remained at essentially the same rate for older age groups over the study period [2], as previously noted [10, 11]. Thus, what has increased is the use of firearms as the preferred method of violence rather than the amount of violence. Nevertheless, the increase in firearms as the method of lethal violence has likely raised homicide rates even more than if other less-lethal methods had been used.

Our results are correlational and raise questions about alternative explanations. It is possible that the increase in use of firearms may be the result of the increasing acquisition of guns that has occurred over the study period [24]. The more guns in circulation, the greater the chances of their use for violent purposes [25, 26], which helps to explain their greater use for violence in the US [27]. Indeed, the trend in gun registrations has followed the same trend as the rate of firearm homicides in young people [24]. However, this explanation only pushes the question back as to why gun acquisition has increased. We have previously shown that violence in TV dramas is associated with increased fear of crime in the US [18], which again suggests a role for entertainment media not only stoking fears of others but also displaying a method for protecting oneself from it, a message commonly employed by the gun industry [28]. Indeed, the gun industry relies on entertainment media as a promotional outlet for its products, as evidenced by its use of product placement in films [29]. The entertainment industry also profits from the popularity of violent programming, which draws audiences and attests to the continued presence of violence in popular TV programming [30]. Thus, the prominent and increasing use of guns for justified purposes on TV, such as by the police or other sympathetic characters, may serve to promote their use by those who may see a need to defend themselves or to harm others.

Another potential explanation is the high level of economic inequality and poverty that has been found to be associated with firearm violence [31, 32]. This is clearly a major factor in homicide rates. However, if changes in economic conditions were responsible for the trends in

firearm homicides, one would expect a jump in firearm violence after the economic collapse of 2007–2009 when poverty rates increased dramatically [33]. There was an increase in youth suicide starting in that year [34], but there is little evidence of change in the use of firearms in homicide at that point in the study period (see Figs 2 and 3).

Income inequality is more difficult to assess [35], but it grew more rapidly from 2000 to 2010 than from 2010 onward [36], which is also inconsistent with the trends in gun homicide. Nevertheless, to check on the possibility that it accounted for the effect of TV gun use, we examined a common measure of inequality, the Gini coefficient [34] as a predictor of the log odds of gun homicide. The index was less highly related to gun homicide than the share of TV gun use for each of the younger age groups (r = .51 vs. .66 for ages 15–24; r = .33 vs. .39 for ages 25–34) and equally related for the oldest age group; (r = .45). In addition, when entered into the equations for each age group, the index was less predictive than TV gun use and never had a coefficient with a *P* value less than .10. Furthermore, the relation for the youngest age group remained reliable (*P* = .013), while it was borderline for the other age groups (*P*'s = .100, .058). It is not altogether clear therefore that economic factors can explain the larger recent increases in youth firearm homicides ages 15–24 and the more gradual increases in other age ranges.

## Limitations

We recognize that a major limitation in this work is the reliance on the correlation between trends in gun violence in the population and on TV. We cannot rule out the influence of other secular changes that may have affected gun violence in both the real world and on TV. More studies at the individual level are needed to test the causal relation between exposure to TV gun violence and use of guns. Indeed, the Institute of Medicine has called for more research on the influence of media entertainment on gun use [37], a topic deserving of future research. We also note that our measures of TV gun violence were related to firearm homicide in all three age groups, with the strongest relation for persons in the 15 to 24 age range. In addition, we only coded any instance of violence within 5-minute segments once even though segments could involve multiple uses of guns or other weapons. We also did not code mere displays of weapons which could also be a source of imitation. Our measure of gun use was restricted to cases in which the weapon caused injury, which was a more stringent and clear index of gun use with the intent to harm. Thus, our measure of gun violence is likely an underestimate of the amount of gun use shown or heard on TV. Finally, we relied on gun homicide as our measure of real-world gun use despite the fact that many more injuries occur from non-fatal gun use [38]. However, homicides are the most sensitive measure of violent injury because they are more carefully tracked by CDC's injury reporting system than non-fatal injuries [39] for which CDC no longer provides annual estimates.

## Conclusions

We recognize that it is unlikely that exposure to TV content is the major source of the longstanding higher rate of gun victimization in youth. However, our findings do add to concerns that the growing presence of guns in entertainment media contributes to their use, an association especially evident in young people. Further research is needed to determine whether exposure to gun violence in entertainment media serves to promote the use of guns, especially by youth.

## Supporting information

**S1 Appendix. US prime-time TV-14 dramas in the sample by number of segments, years coded, and peak rank as listed in variety magazine, 2000–2018.**
(DOCX)

**S1 Dataset.**
(XLSX)

# Acknowledgments

We thank Lauren Hawkins and the team of undergraduate research assistants who conducted the coding of the TV dramas. We also acknowledge the support of the Robert Wood Johnson Foundation.

# Author Contributions

**Conceptualization:** Patrick E. Jamieson, Daniel Romer.

**Data curation:** Patrick E. Jamieson.

**Formal analysis:** Daniel Romer.

**Funding acquisition:** Patrick E. Jamieson.

**Investigation:** Patrick E. Jamieson, Daniel Romer.

**Methodology:** Patrick E. Jamieson, Daniel Romer.

**Project administration:** Patrick E. Jamieson.

**Supervision:** Patrick E. Jamieson, Daniel Romer.

**Validation:** Patrick E. Jamieson, Daniel Romer.

**Visualization:** Daniel Romer.

**Writing – original draft:** Patrick E. Jamieson, Daniel Romer.

**Writing – review & editing:** Patrick E. Jamieson, Daniel Romer.

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
