## [Decision Letter · Decision Letter 0]

15 Dec 2020

PONE-D-20-22320

The Association between the Rise of Gun Violence in Popular US Primetime Television Dramas and Homicides Attributable to Firearms, 2000-2018

PLOS ONE

Dear Dr. Romer,

Thank you for submitting your manuscript to PLOS ONE. After careful consideration, we feel that it has merit but does not fully meet PLOS ONE’s publication criteria as it currently stands. Therefore, we invite you to submit a revised version of the manuscript that addresses the points raised during the review process.

Please address the concerns of reviewer # 2 in your manuscript. Please specifically integrate the following points:

(1) Please make mention of the fairly well-known (or should I say well-accepted?) fact that homicide rates have not rise due to the increase in number and skill of shock-trauma units across the country.

(2) Please italicize the statement, "However, rates of homicide are not the critical outcome to assess the media influence hypothesis."  It is obviously at the core of your paper, and something that you correctly allege is widely  misunderstood.

(3) Please also consider rates of assault in addition to homicide rates.

(4) Please follow up on reviewer #2's suggestion to add peak Nielsen rating for the shows listed in the Appendix.

(5) Abstract, line 5: "ARE attributable"

(6) Please address reviewer #2's disagreement with your criterion of having a gun fired.  ANY display of a gun (and you probably know how frequently that is occurring) should have sufficed!

(7) Please reference the James Bond study (Arch of Pediatr & Adolesc Med 2013; 167(2):195-196), which found that the amount of violence has doubled and the amount of LETHAL violence has tripled in Bond movies over the decades. 

We look forward to receiving your revised manuscript.

Kind regards,

M. Harvey Brenner, PhD

Academic Editor

PLOS ONE

Journal Requirements:

Reviewers' comments:

Reviewer's Responses to Questions

**Comments to the Author**

1. Is the manuscript technically sound, and do the data support the conclusions?

Reviewer #1: Yes

Reviewer #2: Yes

2. Has the statistical analysis been performed appropriately and rigorously? 

Reviewer #1: Yes

Reviewer #2: I Don't Know

3. Have the authors made all data underlying the findings in their manuscript fully available?

Reviewer #1: Yes

Reviewer #2: Yes

4. Is the manuscript presented in an intelligible fashion and written in standard English?

Reviewer #1: Yes

Reviewer #2: Yes

5. Review Comments to the Author

Reviewer #1: This is an excellent manuscript. It deals with the complexity of the theoretical aspects of the media violence/real world violence hypothesis with appropriate measures and data analyses, unlike most articles that claim to find no relation between these two variables. In other words, its no surprise that when one uses inappropriate measures and statistical techniques, one can find null effects. It is refreshing to see a study that uses theoretically appropriate measures and methods.

Here are few suggestions/questions that might be addressed in a final revision.

1. Is it feasible to introduce some statistical controls for other variables that might be related to gun violence? For example, could one add the GINI measure of income inequality to the analysis? I realize that with the small # of data points (i.e., years) that statistically it is risky to add too many controls. And, I also believe that only controls that make theoretical sense should be considered.

2. I think that there is an error in the Figure 4 caption. Specifically, it says "from 2002 to 2018," but shouldn't it say, "from 2000 to 2018"?

3. In the literature review (or discussion), it might be useful to briefly mention the following articles:

Anderson et al., 2010, showed that the effect of violent video games on aggression was not moderated by whether the aggression measure was violent behavior vs. a milder form of aggression. That is, this moderator effect was not significant, suggesting that another form of violent media also increases real world violence.

Anderson, C. A., Shibuya, A., Ihori, N., Swing, E. L., Bushman, B.J., Sakamoto, A., Rothstein, H.R., & Saleem, M. (2010). Violent video game effects on aggression, empathy, and prosocial behavior in Eastern and Western countries. Psychological Bulletin, 136, 151-173.

Dillon & Bushman (2017) showed in an experiment that kids who saw a gun in a movie clip played with a "real" gun hidden in a closed drawer longer and pulled the trigger, relative to kids who saw the same clip but with the gun image cut from the clip.

Dillon KP, Bushman BJ. Effects of exposure to gun violence in movies on children’s interest in real guns. JAMA Pediatr. 2017;171(11):1057-1062. DOI: 10.1001/jamapediatrics.2017.2229

Reviewer #2: This is a very creative and appropriate study from an excellent group of researchers.

A few suggestions and observations:

(1) The authors make no mention of the fairly well-known (or should I say well-accepted?) fact that homicide rates have not rise due to the increase in number and skill of shock-trauma units across the country.

(2) I suggest italicizing the statement, "However, rates of homicide are not the critical outcome to assess the media influence hypothesis." It is obviously at the core of your paper, and something that you correctly allege is widely misunderstood.

(3) I wish you had looked at rates of assault in addition to homicide rates. Perhaps that could be a separate paper?

(4) I would suggest adding peak Nielsen rating for the shows listed in the Appendix.

(5) Abstract, line 5: "ARE attributable"

(6) I'm afraid that I disagree with your criterion of having a gun fired. ANY display of a gun (and you probably know how frequently that is occurring) should have sufficed!

(7) I'm very surprised that the authors did not reference the James Bond study (Arch of Pediatr & Adolesc Med 2013; 167(2):195-196), which found that the amount of violence has doubled and the amount of LETHAL violence has tripled in Bond movies over the decades. I happen to know that the authors wanted to actually count the number of gun rounds expended but were told by sound studios that they couldn't because there were too many in the recent films (personal correspondence).

6. PLOS authors have the option to publish the peer review history of their article (what does this mean?). If published, this will include your full peer review and any attached files.

Reviewer #1: No

Reviewer #2: No

---

## [Author Response · Author response to Decision Letter 0]

17 Jan 2021

Please address the concerns of reviewer # 2 in your manuscript. Please specifically integrate the following points:

(1) Please make mention of the fairly well-known (or should I say well-accepted?) fact that homicide rates have not risen due to the increase in number and skill of shock-trauma units across the country.

We now cite a recent study (p. 1 in Introduction) that indicates that less serious gunshot wounds have resulted in less mortality over time due to care at trauma centers (but that more serious multiple wounds have not). As a result, we take note of the possibility that the decline in gun homicides may have been partly attributable to better care in trauma centers. 

(2) Please italicize the statement, "However, rates of homicide are not the critical outcome to assess the media influence hypothesis." It is obviously at the core of your paper, and something that you correctly allege is widely misunderstood.

Done.

(3) Please also consider rates of assault in addition to homicide rates.

We have now taken note in the Discussion of the fact that CDC no longer provides assault rates for firearm injuries due to concerns about the validity of those rates. In addition, we emphasize that homicides are more reliably tracked than mere assaults, which makes those data more diagnostic for assessing rates of gun use over time.

(4) Please follow up on reviewer #2's suggestion to add peak Nielsen rating for the shows listed in the Appendix.

Done.

(5) Abstract, line 5: "ARE attributable"

Fixed.

(6) Please address reviewer #2’s disagreement with your criterion of having a gun fired. ANY display of a gun (and you probably know how frequently that is occurring) should have sufficed!

We used the criterion of firing a gun and hitting a living target as the metric for the study because it is a more stringent measure of actual gun use and more clearly a measure of intent, which is the meaning of the overall incidence of violence that we coded. There are often occurrences of guns that only involve the sound of gun firing and these are more difficult to code as violence. In addition, if we used any display of a gun, that would not have allowed us to create a proportion of violence measure.

(7) Please reference the James Bond study (Arch of Pediatr & Adolesc Med 2013; 167(2):195-196), which found that the amount of violence has doubled and the amount of LETHAL violence has tripled in Bond movies over the decades.

We’ve added this cite to the paper in the first paragraph of the paper.

1. Is it feasible to introduce some statistical controls for other variables that might be related to gun violence? For example, could one add the GINI measure of income inequality to the analysis? I realize that with the small # of data points (i.e., years) that statistically it is risky to add too many controls. And, I also believe that only controls that make theoretical sense should be considered.

We now discuss the possibility of the Gini coefficient as an alternative explanation in the Discussion. Although this is a standard measure of economic inequality, it does not explain the trend in proportion of gun homicides better than TV gun use and is not a significant predictor in any of the models.

2. I think that there is an error in the Figure 4 caption. Specifically, it says "from 2002 to 2018," but shouldn’t it say, "from 2000 to 2018"?

Fixed.

In the literature review or discussion, it might be useful to briefly mention the following articles.

We now include the study by Dillon & Bushman (first paragraph of Introduction) which specifically involved movies and is directly relevant to our study. The other suggested references refer to video game violence, which are not in the purview of the present study.

We hope these changes address all of the concerns that were expressed and look forward to any further changes you might wish us to consider.

---

## [Decision Letter · Decision Letter 1]

15 Feb 2021

The Association between the Rise of Gun Violence in Popular US Primetime Television Dramas and Homicides Attributable to Firearms, 2000-2018

PONE-D-20-22320R1

Dear Dr. Romer,

We’re pleased to inform you that your manuscript has been judged scientifically suitable for publication and will be formally accepted for publication once it meets all outstanding technical requirements.

Kind regards,

M. Harvey Brenner, PhD

Academic Editor

PLOS ONE

Additional Editor Comments (optional):

Reviewers' comments:

Reviewer's Responses to Questions

**Comments to the Author**

1. If the authors have adequately addressed your comments raised in a previous round of review and you feel that this manuscript is now acceptable for publication, you may indicate that here to bypass the “Comments to the Author” section, enter your conflict of interest statement in the “Confidential to Editor” section, and submit your "Accept" recommendation.

Reviewer #2: All comments have been addressed

2. Is the manuscript technically sound, and do the data support the conclusions?

Reviewer #2: Yes

3. Has the statistical analysis been performed appropriately and rigorously? 

Reviewer #2: Yes

4. Have the authors made all data underlying the findings in their manuscript fully available?

Reviewer #2: Yes

5. Is the manuscript presented in an intelligible fashion and written in standard English?

Reviewer #2: Yes

6. Review Comments to the Author

Reviewer #2: (No Response)

7. PLOS authors have the option to publish the peer review history of their article (what does this mean?). If published, this will include your full peer review and any attached files.

Reviewer #2: No

---

## [Editor Report · Acceptance letter]

22 Feb 2021

PONE-D-20-22320R1 

The Association between the Rise of Gun Violence in Popular US Primetime Television Dramas and Homicides Attributable to Firearms, 2000-2018 

Dear Dr. Romer:

I'm pleased to inform you that your manuscript has been deemed suitable for publication in PLOS ONE. Congratulations! Your manuscript is now with our production department. 

Kind regards, 

on behalf of

Professor M. Harvey Brenner 

Academic Editor

PLOS ONE